# *Stephanoascus ciferrii* Complex: The Current State of Infections and Drug Resistance in Humans

**DOI:** 10.3390/jof10040294

**Published:** 2024-04-18

**Authors:** Terenzio Cosio, Francesca Pica, Carla Fontana, Enrico Salvatore Pistoia, Marco Favaro, Isabel Valsecchi, Nikkia Zarabian, Elena Campione, Françoise Botterel, Roberta Gaziano

**Affiliations:** 1Department of Experimental Medicine, University of Rome Tor Vergata, Via Montpellier 1, 00133 Rome, Italy; pica@uniroma2.it (F.P.); pistoiae@uniroma2.it (E.S.P.); favaro@uniroma2.it (M.F.); roberta.gaziano@uniroma2.it (R.G.); 2Dermatology Unit, Department of Systems Medicine, Tor Vergata University Hospital, 00133 Rome, Italy; elena.campione@uniroma2.it; 3Laboratory of Microbiology and BioBank, National Institute for Infectious Diseases “Lazzaro Spallanzani” I.R.C.C.S., 00149 Rome, Italy; carla.fontana@inmi.it; 4DYNAMYC 7380, Faculté de Santé, Université Paris-Est Créteil (UPEC), 94010 Créteil, France; valsecchiisabel@gmail.com (I.V.); francoise.botterel@aphp.fr (F.B.); 5School of Medicine and Health Sciences, George Washington University, 2300 I St NW, Washington, DC 20052, USA

**Keywords:** *Candida ciferrii*, *Stephanoascus ciferrii* complex, fungal infections, yeast, immunosuppression, antifungal agents, antifungal resistance

## Abstract

In recent years, the incidence of fungal infections in humans has increased dramatically, accompanied by an expansion in the number of species implicated as etiological agents, especially environmental fungi never involved before in human infection. Among fungal pathogens, *Candida* species are the most common opportunistic fungi that can cause local and systemic infections, especially in immunocompromised individuals. *Candida albicans* (*C. albicans*) is the most common causative agent of mucosal and healthcare-associated systemic infections. However, during recent decades, there has been a worrying increase in the number of emerging multi-drug-resistant non-*albicans Candida* (NAC) species, i.e., *C. glabrata*, *C. parapsilosis*, *C. tropicalis*, *C. krusei*, *C. auris*, and *C. ciferrii*. In particular, *Candida ciferrii*, also known as *Stephanoascus ciferrii* or *Trichomonascus ciferrii*, is a heterothallic ascomycete yeast-like fungus that has received attention in recent decades as a cause of local and systemic fungal diseases. Today, the new definition of the *S. ciferrii* complex, which consists of *S. ciferrii*, *Candida allociferrii*, and *Candida mucifera*, was proposed after sequencing the 18S rRNA gene. Currently, the *S. ciferrii* complex is mostly associated with non-severe ear and eye infections, although a few cases of severe candidemia have been reported in immunocompromised individuals. Low susceptibility to currently available antifungal drugs is a rising concern, especially in NAC species. In this regard, a high rate of resistance to azoles and more recently also to echinocandins has emerged in the *S. ciferrii* complex. This review focuses on epidemiological, biological, and clinical aspects of the *S. ciferrii* complex, including its pathogenicity and drug resistance.

## 1. Introduction

In recent years, there has been a significant increase in the number of fungal infections in humans, caused by a wider variety of species, including many environmental fungi that have not been previously associated with human infections. This increase is related to several factors, including iatrogenic immunosuppression [1]; new biological drugs used in the treatment of inflammatory and autoimmune diseases, which target physiological axes involved in host defence mechanisms against fungi [2]; improvement in laboratory diagnostic procedures used for fungal identification [3]; and environmental changes related to rising temperatures that can affect the niche of fungi [4,5,6]. Over the past five years, the LIFE portal has enabled the assessment of the global burden of severe fungal infections for more than 5.7 billion individuals, representing over 80% of the global population [7]. Among fungal pathogens, *Candida* species are the most common cause of mucosal and systemic disease, representing the fourth cause of bloodstream infections in hospitalised patients in the United States and seventh in Europe [8,9]. *Candida albicans* is known as the most common cause of nosocomial invasive candidiasis. Nevertheless, in the past ten years, an increased prevalence of non-*albicans Candida* species, including *C. parapsilosis*, *C. glabrata*, *C. krusei*, *C. tropicalis*, *C. sojae*, *C. auris*, and *C. ciferrii*, has been reported [10,11,12]. Among NAC spp., *Candida ciferrii*, named in memory of the Italian mycologist Raffaele Ciferri, also known as *S. ciferrii* or *Trichomonascus ciferrii*, has gained our attention in the past years as a multi-drug-resistant pathogen able to cause both local and systemic infections [13,14,15]. The *C. ciferrii* species is a heterothallic ascomycete yeast-like fungus. Although the first cases of *C. ciferrii* infection were described in 1960 by Kreger-Van et al. [15], in 1983, Furman and Ahearn [16] reported five clinical cases of suspected fungal infection where *C. ciferrii* was isolated, but he stated to have insufficient evidence to establish that *C. ciferrii* was a pathogenic yeast. However, as reported by Furman and Ahearn, clinical laboratories should be alerted to their diagnostic properties since this species is an emerging fungal pathogen currently detected in human clinical samples [16]. A few case reports of different types of human fungal diseases due to *S. ciferrii* worldwide have gained particular attention on this yeast as a real pathogen [17,18,19,20,21,22]. Just over 20 years ago, thanks to the 18 S rRNA sequencing technique, it was understood that at least three different species, with significant genetic similarity, namely *S. ciferrii*, *Candida allociferrii*, and *Candida mucifera*, could have been considered together within a single group, the so-called “*S. ciferri* complex” [23]. However, in the relatively few clinical reports available in the specific literature [17,18,19,20,21,22], this grouping was often not taken into account so that in various types of clinically documented fungal infections, species investigations were not performed [21,22,23,24] and the isolated fungal agent was indicated using various and non-univocal terminology. Identifying the *S. ciferrii* complex at the species level is of great importance in epidemiological studies and for the management of empirical antifungal therapy [25]. However, limiting data concerning the epidemiological, etiologic, and clinical aspects of the *S. ciferrii* complex are reported in the literature.

This review aims to provide important information on the epidemiology, biology, and clinical features of the *S. ciferrii* complex in order to improve our knowledge and clinical management of *S. ciferrii*-related infections in view of its multi-drug resistance.

## 2. Materials and Methods

### 2.1. Search Strategy

A thorough investigation was conducted across various databases considering different types of published articles from January 1965 to December 2023. In detail, Cochrane Central Register of Controlled Trials, MEDLINE, Embase, US National Institutes of Health Ongoing Trials Register, NIHR Clinical Research Network Portfolio Database, and the World Health Organization International Clinical Trials Registry Platform were examined. The utilised search terms were “*Candida ciferrii*”, “*Stephanoascus ciferrii*”, “*Trichomonascus ciferrii*”, “*Stephanoascus ciferrii* complex”, “*Candida mucifera*”, and “*Sporothrix catenate*”.

### 2.2. Inclusion Criteria

To investigate human infections due to the *S. ciferrii* complex, in all the studies where in addition to *S. ciferrii* other fungal pathogens were included, only data related to the *S. ciferrii* complex were considered. All human studies were included, with no age-, sex-, ethnicity-, or type-of-study-related restriction. Case reports and case series describing human infections due to the *S. ciferrii* complex that have not yet been included in reviews or trials were also included.

### 2.3. Exclusion Criteria

The targeted intervention excluded the analyses of other infections that were not correlated with the *S. ciferrii* complex.

### 2.4. Data Collection

Charts and tables were reviewed for clinical details. Data collected included demographics, location, risk factors, affected area, causative species, diagnostic investigations, treatment, and outcome.

## 3. Results

### 3.1. S. ciferrii Complex: A Real Fungal Pathogen

Eighty-one articles or trials regarding *S. ciferrii* complex infections were identified through this quantitative research. Forty-three articles were excluded after applying the exclusion criteria. Among the thirty-eight articles or trials eligible for evaluation, ten were excluded as duplicated, while six were excluded after reading the abstract or full text. Twenty-two articles or trials were evaluated in this review (Appendix A). The results of our research are summarised in Table 1.

### 3.2. History

*Candida ciferrii* was isolated for the first time from animals in 1962 as reported by Kreger-Van [15]. It was reported that, over a two-year period, The Yeast Division at Delft had received four yeast strains to be identified: one of them was from Dr. Seeliger (Bonn, Germany) in 1961, but its origin was not specified; two were from a cow’s neck and from a wooden pole in a cow shed, respectively, by Dr. Klot; and the fourth was by Dr. Swierstra (Utrecht, the Netherlands) and detected from the throat of a pig, in 1963 [15]. Twenty years later, in 1983, suspected human infections due to *C. ciferrii* were reported in America by Furman and Ahearn [16]; however, the researchers reported insufficient evidence to confirm *C. ciferrii* as a pathogenic yeast (Figure 1). Then, from 1990, *C. ciferrii* was isolated from patients in Europe (Angers, France) by De Gentile et al. [13] and, from 2000, in Asia (China and India), paving the way for considering *C. ciferrii* as a pathogenic microorganism. In 2002, in light of the microbial complex definition [21], Ueda-Nishimura and Mikata classified *S. ciferrii* into three categories and introduced the term *S. ciferrii* complex, encompassing *Stefanoascus ciferrii*, *Candida allociferrii*, and *Candida mucifera*, through the analysis of the 18S rRNA gene sequence [23].

### 3.3. Morphology and Identification of S. ciferrii Complex

*Stephanoascus ciferrii* (also called *Candida ciferrii* or *Trichomonascus ciferrii*) is a heterothallic ascomycete yeast-like fungus, which is a teleomorph of *Candida ciferrii*. Furthermore, *Candida ciferrii* was classified as an ascomycete because it is not able to give a colour reaction with diazonium blue B and to divide by binary fission and fails to yield an ascogenous state in either single or mixed cultures [24]. Concerning the growth and morphological characteristics of the *S. ciferrii* complex, the fungus grows on Sabouraud dextrose agar (SDA) at different temperatures (22 °C, 27 °C, and 37 °C) after one week. Colonies are typically small, round, and milky-white- or cream-coloured, and can be hard, raised, and wrinkled or smooth [13,16,25,42]. Moreover, the *S. ciferrii* complex can grow on SDA with 0.1% cycloheximide, SDA with olive oil overlay, or 10% NaCl [36]. After 72 h, colonies appear to be embedded into the agar with a hard texture. On the chromogenic culture media like CHROMagar™ (Candida CHROMagar™, Paris, France), colonies also grow well, are regular round with a blue centre and white edge, and become rough with gyrus-like grooves and irregular edges in 7-day cultures [25]. The chromogenic HiCrome™ Candida Differential Agar shows dry blue-white, wrinkled colonies [36], whereas a tetrazolium reduction medium (TTZ) shows pink to white dry, rough, and wrinkled colonies. The latter on potato dextrose agar (PDA), oatmeal agar (OMA), and corn meal agar (CMA) are rough, dry, white, and wrinkled [36]. The morphological features of the *S. ciferrii* complex colonies are summarised in Table 2.

Microscopically, the ascus of the *S. ciferrii* complex contains two helmet-to-hat-shaped ascospores, chlamydospore-like cells, and blastoconidia on denticles [16]. The mycelium is abundant, branched, septate, with small ramified chains of oval blastospores, variable in size, and arranged alongside the hyphae [13]. Contrary to *C. albicans*, the germ tube test is negative at 37 °C and both the chromogenic media, CHROMagar *Candida*, and the auxanogram seem to be not useful for the identification of the fungus [43]. Regarding the biochemical profile, it can assimilate glucose, galactose, saccharose, maltose, raffinose, D-xylose, cellobiose, trehalose, L-arabinose, D-ribose, inositol, erythritol, ribitol, D-mannitol, citric acid, succinic acid, allantoin, and adenine [16,24], but not the raffinose [24]. It does not ferment galactose, lactose, maltose, sucrose, and raffinose. Further, the splitting of arbutin is positive [15]. It should be noted that both the macroscopic and microscopic characteristics of the three species of the *S. ciferrii* complex are similar, making it hard to distinguish them based on their morphology. Moreover, based on morphological and biochemical characteristics, *C. ciferrii* can be misidentified with other fungi such as *Cryptococcus neoformans* and *Cryptococcus laurentii*. Warren et al. [14] reported a case of *C. laurentii* infection in a 24-year-old woman with precipitous respiratory failure after lung transplantation that was misidentified with non-pathogenic *S. ciferrii* by the Vitek 2 system (BioMérieux, Marcy l’Etoile, France) [14] (Table 3). Also, Kuram et al. [44] reported two cases of suspected systemic infection due to *Blastobotrys serpentis* and *Blastobotrys proliferans* misidentified as *S. ciferrii* in a preterm newborn and a rhabdomyosarcoma patient, respectively, by using a Vitek 2 yeast ID system. Moreover, Al-Haqqan et al. [45] reported a case of *Candida blankii* bloodstream infection identified as *S. ciferrii* by the Vitek2 yeast identification system with an 89% probability. Also, in the study, the matrix-assisted laser desorption/ionisation time-of-flight mass spectrometry (MALDI-TOF MS) was not able to identify *C. blankii*, which was successfully identified by an Internal Transcribed Spacer (ITS) PCR analysis (Table 3). MALDI-TOF MS is one of the most commonly used proteomic techniques in the identification of microbial species [46]. Marklein et al. [47] evaluated the use of MALDI-TOF MS technology for the rapid and reliable identification of clinical yeast isolates. *Candida ciferrii* species has been currently included in commercial MALDI-TOF MS databases; however, this proteomic system is not able to distinguish between *C. allociferrii* and *C. mucifera* [25]. Sathi et al. [34] in a study performed in 2021, in a tertiary care hospital in Bangladesh, to determine the prevalence of *Candida* spp. clinical isolates and their susceptibility to antifungal drugs, isolated 12 *C. ciferrii* strains from different specimens. Conventional culture methods failed to identify *C. ciferrii* isolates that were identified by PCR, followed by Restriction Fragment Length Polymorphism (RFLP) tests [34]. To date, the MALDI-TOF MS technology appears to be the fastest method for *S. ciferrii* identification at the genus level, but molecular tests have proven to be a useful tool for the identification of the *S. ciferrii* complex at the species level. Despite the MALDI-TOF MS technology including *S. ciferrii*, a case report from Roehmel et al. [48] highlighted that *Arxula adeninivorans*, a yeast closely related to *S. ciferrii*, can be misdiagnosed as *S. ciferrii*. Balada-Llasat et al. [49] proposed a multiplex PCR xTAG fungal ASR assay for detecting clinically relevant *Candida* spp., *Cryptococcus neoformans*, *Histoplasma capsulatum*, and *Blastomyces dermatitidis* from blood cultures. One blood culture that was spiked with *Candida ciferrii* was determined to be *Cryptococcus neoformans* by the multiplex xTAG fungal ASR assay. While repeat testing by xTAG blood culturing (and extracted nucleic acid) yielded the same result, the reference laboratory confirmed the culture identity of the isolate as *C. ciferrii* by DNA sequencing [49]. Overall, this study confirms that molecular tests, and specifically the sequencing of ITS regions, allow us to identify the *S. ciferrii* complex even at the species level (Table 3).

In recent years, there has been a significant increase in the number of studies utilising whole-genome sequencing to analyse pathogenic genomic determinants in fungi [50,51]. Mixao et al. [22] conducted a genome sequencing of *T. ciferrii* and found that the genome size is approximately 19 Mb and highly homozygous, with 0.09 heterozygous SNPs/kb, based on the same 27-mer counts. Additionally, they reconstructed the complete collection of gene evolutionary histories [52] for *T. ciferrii*, comparing it with twenty-six other species. They identified 1217 genes specific to *T. ciferrii*, out of which only 391 had homologs in species not considered for phylome reconstruction. Interestingly, the genes specifically duplicated in *T. ciferrii* appeared to be enriched in transmembrane transport activities and oxidoreductase activity. From their study, they concluded that *T. ciferrii* is closely related to *Yarrowia* (*Y*) *lipolytica*. However, when examining the MAT locus of *T. ciferrii*, Mixao et al. [22] observed that unlike *Y. lipolytica*, which has both MAT a and MAT alpha alleles [53], *T. ciferrii* only possesses the MAT alpha allele. It is important to note that although there is a protein-coding gene in place of MAT alpha2, this protein does not have any homologs, and therefore only MAT alpha1 could be identified in the *T. ciferrii* genome.

### 3.4. Epidemiology

The *Stephanoascus ciferrii* complex has been globally isolated from human samples and reported in all continents except Australia and Antarctica, confining the presence of the *S. ciferrii* complex to the northern hemisphere, with the only exception of one case in Brazil [40]. In this review, the world distribution of the 92 *S. ciferrii* complex clinical isolates is reported in Figure 2. The majority of *S. ciferrii* complex strains have been isolated in Asia (66%), followed by Europe (25%) and America (9%). In Asia, China is the country with the highest isolation rate (61%). In Europe, France ranks first (78% of cases), whereas in America, it is Mexico (25%). The report illustrating the cases registered in each state is shown in Appendix A. Furthermore, considering the identification period, a bimodal trend can be noted, in which most of the cases were described between 1983 and 1995, and then between 2018 and 2021, respectively, in Europe (France) and Asia (China), as reported in Appendix A. Data from the ITS PCR analysis are still limited, and only Soki et al. [17], in 2015, described a *C. allociferrii* infection by using gene sequencing, followed by Guo et al. [25], in 2021, who classified the 32 yeast strains isolated in their study as the *S. ciferrii* complex at the species level. Based on the alignment results, 16 strains of *S. ciferrii* (50%), 10 strains of *C. allociferrii* (31.25%), and 6 strains of *C. mucifera* (18.75%) have been reported [25].

Interestingly, *C. mucifera* has only been described in human samples in China, while *C. allociferrii* has been reported in China and Japan.

### 3.5. Clinical Features

#### 3.5.1. Auricular Infections

Most fungal ear infections are caused by *Aspergillus* spp., which represent the most frequent etiologic agent, followed by *Candida* spp. [54]. In recent years, there has been an increasing incidence of otomycosis due to NAC species with high resistance and recurrence. Among these, as reported in a recent study by Alam et al. [54], *C. parapsilosis* is the predominant species followed by *C. tropicalis* and *C. famata*. However, rare species of emerging drug-resistant *C. ciferrii* have also been correlated with ear infections. It should be noted that among the 92 documented cases of *S. ciferrii* complex human infections, 41% are represented by otomycosis, which include chronic suppurative otitis media (CSOM) and granular myringitis. The first suspected case of CSOM was reported by Furman et al. [16] in 1983, in a 25-year-old male patient presenting with oesophageal pain, headache, and pressure in the ears. Otomycosis was diagnosed and the patient was treated with local 1% hydrocortisone and 2.0% acetic acid solution with complete recovery. But it is only in 2019 that the first case of *S. ciferrii*-complex-related CSOM was identified in India. Romald et al. [36] reported a case of CSOM from the aural discharge after mastoidectomy in a female immunocompetent patient, which was successfully treated with oral voriconazole and topical clotrimazole. Guo et al. [25] reported the majority of *S. ciferrii*-complex-related CSOM from China in 2021. In that study, among the 32 patients with CSOM, a parallel prevalence between males, 53.22% (17/32), and females, 46.88% (15/32), was observed with no statistically significant difference. The patients’ ages ranged from 20 to 70 years, with a median age of 36 years. Among the patients, the *S. ciferrii* complex was found in 31.25% (10/32) of those under 30 years old, 59.38% (19/32) of patients aged between 30 and 60 years, and 9.37% (3/32) of individuals over 60 years old. Through the sequence analysis of the rRNA gene ITS regions, sixteen strains of *S. ciferrii* (50%), ten strains of *C. allociferrii* (31.25%), and six strains of *C. mucifera* (18.75%) were identified. Considering that CSOM is a major cause of hearing impairment worldwide, in all cases of CSOM unresponsive to local antibiotics, superimposed fungal infection should be suspected [55]. Despite that most of the auricular infections caused by the *S. ciferrii* complex are classified as CSOM, Chan et al. [39], in 2016, reported four patients (three males and one female), with a mean age of 38 years, with *S. ciferrii*-associated granular myringitis but they had no additional comorbidities except for one patient with nasopharyngeal carcinoma, diabetes mellitus, and hyperlipidaemia. Their symptoms included obstructive sensation and/or pruritus in their ears, pustular otorrhoea, and otalgia. All the patients responded to topical azole treatment. Considering that granular myringitis is an uncommon ear disease [53], characterised by the de-epithelialisation of the lateral squamous layer of the tympanic membrane and replacement with granulation tissue in the absence of middle-ear disease, in most cases, it causes a chronic disorder that is frequently misdiagnosed as CSOM due to concomitant bacterial complications [56]. Consistently, although bullous myringitis is generally thought to be caused predominantly by viruses, several bacterial strains have been associated with this clinical manifestation [57].

#### 3.5.2. Infections of the Skin and Its Appendages

Superficial mycoses of skin and skin appendages are the most common human fungal infections. They can be caused by dermatophytes and non-dermatophyte moulds and yeasts [58]. *Candida* species have been found in a relevant number of cases, either as the primary pathogens or in combination with dermatophytes and other fungal types [53,58]. By our analysis, we found that twenty-four percent of *S. ciferrii* complex clinical isolates were from infections of the skin and skin appendages, with a predominance of the last types of infections, i.e., onychomycosis or toenail onyxes [13,16,26]. Moreover, only 5/26 (19%) patients presented concomitant skin ulcers. The presence of skin ulcerative lesions highlights a possible role of cutaneous hypoperfusion in the onset of *S. ciferrii* complex infection, especially in conditions of hypoxia or peripheral vascular failure. Assessing *Candida* skin infections in patients with peripheral vascular insufficiency is extremely important to establish the *S. ciferrii* complex as a skin colonizer that under favourable conditions can lead to overt disease. Interestingly, 18 cases (62%) were reported from two studies conducted in France in 1991 and 1995 [13,26], while 4 cases have been reported in America, 3 in 1983 [16] and the last one in 2022, which was complicated by systemic spread and exitus [33]. The lack of data from 1983 to 1995 do not allow us to identify specific risk factors and patients’ outcomes. In this regard, more knowledge is needed about the host–microbiome crosstalk in the genesis and outcome of complicated chronic wounds, with the view of developing more appropriate therapeutic approaches [59].

#### 3.5.3. Ocular Infections

Fungi represent an important cause of microbial ocular infections and, in some geographical areas, in tropical and developing nations, they are significant contributors to vision loss, leading to serious eye infections like keratitis and endophthalmitis [60]. Globally, ocular trauma or surgery are the most important predisposing cause of exogenous ocular mycoses, with the species of *Fusarium*, *Aspergillus*, and *Candida* being the most frequently isolated microorganisms [61,62]. Among *Candida* species, the *S. ciferrii* complex was isolated from ocular specimens in seven cases, five of them associated with chronic postoperative endophthalmitis [20,31], one from endophthalmitis with an intraorbital abscess in a patient after the enucleation of choroidal melanoma [32], and the last one from an intraorbital abscess in a Japanese patient [17]. All patients recovered from these ocular infections. In addition, two other cases of ocular diseases related to the *S. ciferrii* complex but without other specifications have been reported by Sathi et al. [34]. Most cases recurred in India, in female patients after iatrogenic procedures, supporting the hypothesis that ocular infections may be associated with hospital procedures or a nosocomial environment.

#### 3.5.4. Candidemia

Despite adequate treatment, candidemia remains one of the main causes of nosocomial bloodstream infections (BSIs) with an increasing incidence detected in Europe, presumably due to fungal resistance caused by the overuse of first-line antifungals, and an attributable mortality of 30–40% [61,62,63,64,65]. Consistently, between 2000 and 2019, the incidence of candidemia in Europe increased alarmingly, rising from 2.2 to 3.2 cases per 100,000 inhabitants per year. Together with the increased number of immunosuppressed people, which is growing due to innovative and successful therapeutic advances, it is known that one of the driving forces behind fungal resistance is the overuse of first-line antifungal agents [66,67]. This is more common in high-risk environments, where antifungals are routinely used for prophylaxis and/or empirical treatment [68]. As a result, rising trends in azole and echinocandin resistance can severely impair candidemia management due to limited alternative therapeutic options. It is therefore critical to be aware of both emerging resistance and species distribution in each single clinical setting to tailor a prudent and appropriate use of antimycotic agents [69]. Non-*albicans Candida* species are increasingly found as causative agents of human bloodstream infections. Among these, the *S. ciferrii* complex was recovered from the blood of 18 patients with invasive candidiasis. In a study conducted in Taiwan by Cheng MF et al. [28] in which *Candida* species were isolated from blood samples between 1996 and 1999, only one strain of *C. ciferrii* was detected (0.3%) among 383 candidemia cases. Capoor et al. [41] described the first fatal case of bloodstream infection in India in a severely immunocompromised woman affected by AML. Another case of systemic *S. ciferrii* complex infection in a 62-year-old immunocompromised patient with AML was also reported by Gunsilius et al. [27], with complete recovery of the patient. Moreover, cases of candidemia associated with the host immunosuppression were described in a patient with Crohn’s and *Mycobacterium bovis* disease [30] and in a debilitated child with Downs syndrome and cerebral palsy who had undergone a broad-spectrum antimicrobial therapy. In this case, the fungus presented a multi-drug-resistant profile to AmB, fluconazole, and echinocandins and the patient expired [19]. Other cases of non-fatal candidemia caused by *C. ciferri* were reported in Turkey [37] and Asia (Taiwan) [28]. In the latter case, the *C. ciferrii* isolate was resistant to AmB and exhibited a dose-dependent susceptibility to fluconazole (32 μg/mL) [28]. Thus, the host’s immunological status, the hospitalisation, and the exposure to wide-spectrum antimicrobial therapy are all important risk factors for invasive infection due to the *S. ciferrii* complex as well.

#### 3.5.5. Vaginal Infections

Vulvovaginal candidiasis (VVC) is a common infection affecting 2/3 of reproductive-age women worldwide [70]. Although *C. albicans* is considered the most common agent of VVC, other NAC species have been recently identified as causative agents of these infections. Among these, the *S. ciferrii* complex has been isolated from VVC. A study conducted by Sathi et al. [34] demonstrated that among 175 patients with suspected VVC, 52 (29.7%) were positive for *Candida* species. In detail, 34 (65.0%) were *C. albicans* isolates and 18 (35.0%) were NAC. Although *C. glabrata*, *C. tropicalis*, and *C. parapsilosis* were found to be the most common NAC species associated with VVC, one *C. ciferrii* strain was isolated from a patient. The first cases in Malaysia were reported by Ng et al. [35] in a retrospective study conducted from 2000 to 2013, where the authors isolated three *S. ciferrii* complex strains from vaginal swabs collected only in 2000. It is worth noting that infections caused by NAC have become a constant public health threat due to the emergence of drug resistance to commonly used antifungal agents in these fungal pathogens. Therefore, the increasing incidence of NAC associated with vaginitis and high resistance to antifungals should be carefully considered in gynaecology wards and outpatient clinics.

#### 3.5.6. Pulmonary Infections

Chronic obstructive pulmonary disease (COPD) is a chronic inflammatory lung disease that causes airflow obstruction in the lungs. Intensive care unit (ICU) admission for patients with COPD is generally due to infections caused by bacterial or viral pathogens [71]. However, fungal pulmonary infections may also occur in different immunosuppression conditions, such as diabetes mellitus, prolonged high-dose glucocorticoid (GC) use, transplant recipients, and in patients infected with untreated human immunodeficiency virus. Saha et al. [29] (2013) reported a rare case of *S. ciferrii* complex pulmonary infection, occurring in India, in a 55-year-old immunocompromised female patient, being diabetic, a smoker, and hospitalised for pneumonia. She was treated with intravenous liposomal AmB (150 mg daily) and after 4 days switched to oral fluconazole (150 mg daily), which led to her recovery. No data were reported about the susceptibility of this fluconazole-sensitive isolated strain to other antifungal drugs [29]. In addition, a *C. ciferrii*-related septic pulmonary embolism (SPE) was reported by Papìla et al. [38] in a 70-year-old female patient with type 2 diabetes mellitus, which was successfully treated with intravenous anidulafungin, being discontinued to oral fluconazole. In this regard, it is worth mentioning that SPE is a rare disease characterised by fever and imaging findings of multiple nodules or local infiltrates, with or without cavitation, that have been documented in high-risk groups of patients, such as intravenous drug users or patients with intravascular indwelling catheters. The microbial aetiology in these patients includes methicillin-resistant *S. aureus* (MRSA), methicillin-sensitive *S. aureus* (MSSA), *Fusobacteria*, *Klebsiella pneumoniae*, and, more rarely, *Candida* and viridans streptococci as reported by Ye et al. [72] in a systematic review. Curiously, fungal SPE is a rare condition but we must consider the possibility of a superinfection as a consequence of heavy antimicrobial therapies that could lead to a fungal SPE. In our cases, no previous antibiotic treatments were performed; thus, a *S. ciferrii*-complex-related SPE could suggest a hidden infection in another location. All the human infections are resumed in Figure 3.

### 3.6. Treatment and Resistance

The analysis of available literature data shows that 75% of *S. ciferrii* complex isolates are resistant to azoles, and specifically 49% show resistance to fluconazole, which represents the first line of treatment for both systemic and local candidiasis, and is also used to prevent invasive candidiasis in ICUs [73]. However, fluconazole exerts only fungistatic activity against *Candida* spp. that in long-term treatment can develop fluconazole resistance [74]. It is crucial to trace the susceptibility profile of *S. ciferrii* complex isolates as the resistance rate leads to therapeutic failure in the case of empirical therapy. One hundred percent (41/41) of clinical isolates from patients with ear infections showed resistance to fluconazole, followed by 71% of blood infections and 64% of skin and skin appendage disorders. Fluconazole remains the first-choice drug for treating candidiasis, but the increased number of fluconazole-resistant strains made it necessary to look for other antifungal drugs. In this regard, itraconazole, a triazole derivative, exhibits a comparatively wider range of effectiveness against both *Candida* and *Aspergillus* species [75].

However, recently, several *Candida* species have shown increasing resistance to itraconazole in clinical practice [75]. Interestingly, a *S. ciferrii* complex presenting itraconazole resistance was identified in 54% of skin and skin appendage disorders, especially onychomycosis [26], and in 41% of blood infection [34]. Similarly, all cases of *C. ciferrii* resistant to isoconazole, which is generally used in the treatment of superficial skin and vaginal infections, were recovered from patients with onychomycosis in France [26]. Interestingly, no itraconazole resistance has been described in strains from ocular diseases (Table 4).

Even though azoles are widely used in clinics, some patients cannot be treated with these antifungals because they require other medications that have unfavourable drug–drug interactions (DDIs) with azoles or are infected with an azole-resistant isolate of *Candida* spp. or are intolerant to azole therapy. In this view, caspofungin was the first echinocandin antifungal agent to gain FDA approval for use in the USA in 2003 [76]. The IDSA (Infectious Diseases Society of America) recommends echinocandins as the first-line treatment for candidemia, rather than fluconazole, due to their fungicidal activity. Interestingly, in 2023, the FDA approved intravenous rezafungin injection (Rezzayo, Cidara Therapeutics/Melinta Therapeutics), a novel echinocandin, for the treatment of candidemia in adults ≥ 18 years old with limited or no alternative treatment options [77]. Unfortunately, the increased use of echinocandins has been associated with the emergence of resistance to these drugs, particularly in *C. albicans* and *C. glabrata* [78]. Resistance to echinocandins in *Candida* species arises due to specific mutations found in the hotspot regions of the *FKS* gene, which is responsible for encoding the 1,3-beta-D-glucan synthase [79]. Since 2011, echinocandin resistance has been described in clinical isolates of the *S. ciferrii* complex. Agin et al. [19] reported a case of candidemia by *C. ciferrii* in an immunocompromised 8-year-old child. The isolated strain was resistant to AmB (MIC > 1 μg/mL), fluconazole, (MIC ≥ 64 μg/mL), caspofungin (MIC ≥ 32 μg/mL), and anidulafungin (MIC ≥ 32 μg/mL) but sensitive to voriconazole (MIC ≤ 0.12 μg/mL). Ten years later, Guo et al. in 2021 [25], in their case series of CSOM, due to the *S. ciferrii* complex, reported that *C. mucifera* had higher MIC values to echinocandins (anidulafungin, micafungin, and caspofungin) than *C. allociferrii*. Moreover, *C. mucifera* was highly resistant to AmB when compared to *C. ciferrii*, even if the last one displayed higher MICs for flucytosine than *C. allociferrii* and *C. mucifera*. The distribution of resistant isolates of the *S. ciferrii* complex by specimens is reported in Figure 4.

Polyenes were the first commercially available antifungal drugs after griseofulvin and since them, more than 200 polyene antifungals have been discovered. Among them, AmB, nystatin, and natamycin are the most used drugs for the treatment of fungal infections [80]. Currently, three polyenes are still widely utilised for therapeutic purposes. These include AmB, which is used for systemic mycoses, nystatin, which is effective against mucosal infections, and natamycin, which is used for ophthalmic infections [81,82]. However, various studies indicate that the AmB effectiveness in treating systemic mycoses caused by *Aspergillus terreus*, *Scedosporium* spp. [82], and *Candida auris* [83] is sometimes limited by inherited or acquired antifungal resistance.

According to the IDSA [63] and European Confederation of Medical Mycology [84] (ECMM) guidelines, AmB is still recommended as the first-line treatment for severe cryptococcosis (often in combination with flucytosine), disseminated histoplasmosis, and mucormycosis, while it remains an alternative drug for other infections. Furthermore, AmB has been recommended in the prophylaxis of invasive *Candida* and *Aspergillus* infections in solid-organ-transplant recipients and in patients receiving immunosuppressive treatment, respectively [74]. Among the 15 isolates of the *S. ciferrii* complex resistant to AmB, 43% were associated with endophthalmitis (especially in India) and 41% with blood infections (most from Bangladesh). In contrast, the isolates from the ear and skin infections exhibited a low AmB resistance rate. Flucytosine, also identified as 5-fluorocytosine (5-FC), was initially approved as an antifungal medication in the 1970s. Despite this, its therapeutic significance has been underestimated for many years, and its accessibility worldwide remains restricted [85]. The level of primary flucytosine resistance differs across various fungal species. For instance, the reported resistance rates for *C. albicans*, *Cryptococcus neoformans*, and NAC species are 7–8%, 1–2%, and 22% isolates, respectively [86]. Resistance to 5-FC has been found in *C. auris*, an emerging multidrug-resistant NAC species. In detail, Chowdhary et al. showed that 47% of 15 *C. auris* isolates were resistant to 5-FC, exhibiting MICs ranging from 0.25 to 64 µg/mL [87]. The data reported in the literature demonstrate that the strains showing resistance to 5-FC were all isolated in China from auricular samples (94.5%), and those responsible for cutaneous infections (4.5%) in France, especially in younger patients (<38 years old) (Figure 5).

Evaluating the resistance rate of the *S. ciferrii* complex on geographical distribution, the most cases of azole resistance are described in China (54%), followed by France (19%) and Bangladesh (16%), in parallel with resistance to 5-FU observed in China (81%) and India (14%) and the unique case is reported in France. Regarding AmB resistance, Bangladesh (50%) and China (17%) are the countries with the highest prevalence. Regarding echinocandins, in Turkey, a case of anidulafungin and caspofungin resistance (both MIC ≥ 32 μg/mL) has been reported while *C. mucifera* isolated strains in China presented a MIC value ranging from 0.03 to 0.25 μg/mL for micafungin and from 0.015 to 0.03 for anidulafungin and caspofungin [25] (Appendix A). Moreover, in 2019, Pérez-Hansen et al. [88] reported the values of eight fluconazole-resistant *C. ciferrii* strains for anidulafungin, micafungin, and caspofungin, i.e., from 0.03 to 0.125, from 0.06 to 0.125, and from 0.5 to 1 mg/mL, respectively. To date, no EUCAST and ECOFF cut-offs are available to determine whether a *S. ciferrii* complex strain can be considered resistant at a MIC ≥ 0.03 μg/mL. Further studies are needed to establish correct cut-offs for these NAC species.

Interestingly, considering the phylogenetic analyses of *S. ciferrii*, two closely related yeasts, *Yarrowia* (*Y.*) *lypolityica* and *Arxula* (*A.*) *adeninivorans*, share the same resistance profile shown in the *S. ciferrii* complex. Alvarez-Urıa et al. [89] have described a fungemia due to *A. adeninivorans* that presents high MIC (mg/L) values for fluconazole (64), voriconazole (1), posaconazole (2), amphotericin B (2), and anidulafungin (0.5). However, no proposed mechanisms for the resistances have been described. *Y. lipolytica* is an opportunistic pathogen that can cause invasive candidiasis [90]. In 2020, Stavrou et al. [91] demonstrated that twenty-seven *Y. lipolytica* isolates had high MICs for FLC, ITC, and POS. In 2003, Lavergne et al. [92] reported a case of fungemia due to *Y. lipolytica* resistant to fluconazole (MIC ≥ 256 mg/mL). After genome sequencing, they described the A395T mutation in the *ERG11* gene, previously described in fluconazole-resistant *Candida* strains, which can explain the high MIC found in this clinical isolate. In line with the publication of Lavergne et al. [92], Yu et al. [93] reported four additional azole-resistant *Y. lipolytica* clinical strains presenting the A395T mutation in the *ERG11* gene. Despite the rarity of *Y. lipolytica* infections, it is important to emphasize that there has been an emergence of acquired azole cross-resistance. Based on this evidence, we suspect that the high rate of azole resistance described in the *S. ciferrii* complex could be due to the A395T mutation in the *ERG11* gene, but further studies must be performed to clarify this aspect. As well as for azoles, increasing rates of echinocandin resistance have been described in recent years [94,95]. While reported cases of resistance to echinocandins in several *Candida* spp. are widely described, few data are available on *Y. lypolityica* and *A. adeninivorans*. Marcos-Zambrano et al. [96] reported a case of *A. adeninivorans* resistant to azoles, polyenes, and echinocandins. Moreover, based on the previous study of Díaz-García et al. [97], where the isolated strain of *A. adeninivorans* presented MIC ≥ 8 mg/L for the echinocandins, they stated that *A. adeninivorans* is intrinsically resistant to echinocandin. Data regarding the resistance of *Y. lipolytica* against echinocandins are controversial. Stavrou et al. [91] demonstrated that among twenty-seven *Y. lipolytica* isolates, 74.1% were resistant to echinocandins. In contrast, Yu et al. [93] have not reported resistance strains of *Y. lipolytica* in their retrospective study, and the use of echinocandins in bloodstream infections due to *Y. lipolytica* had a favourable outcome [98]. Based on the literature, no data regarding the resistance mechanism for echinocandins are described for the *S. ciferrii* complex as well as for *Y. lipolytica*, but elevated echinocandin MICs have been associated with several single amino acid substitutions caused by mutations in specific “hot spot” regions of the well-conserved target genes *FKS1* of *Y. lipolytica* [99]. Despite the low incidence of described *S. ciferrii* complex echinocandin resistance [19], appropriate antifungal stewardship should be actualised to avoid the emergence of resistance and further studies on peculiar targets such as Heat Shock Protein (HSP) 90 [100] should be performed to clarify the resistance in the *S. ciferrii* complex.

### 3.7. Virulence Factors

Although *Candida* spp. are recognised as commensals, colonising the mucosal surfaces asymptomatically, they can cause significant discomfort and sometimes even fatal infections. Several virulence factors facilitate the transition of *Candida* spp. from the commensal to pathogen status, including the adherence to host tissues and medical devices, biofilm formation, secretion of extracellular hydrolytic enzymes (e.g., proteases, phospholipases, haemolysins, esterases, and phosphatases), toxins, and phenotypic switching [101]. Typical manifestations of infections induced by NAC spp. are usually clinically indistinguishable; nevertheless, several of them may either acquire resistance over time or have innate resistance against routine antifungals or sometimes both [102]. In the case of the *S. ciferrii* complex, several related infections have been described, paving the way to consider it as a pathogenic yeast, especially for patients with comorbidities or immunosuppression states [101]. In particular, Vieira de Melo et al. [40] evaluated the virulence factors of *C. ciferrii* (Figure 6). The adherence to buccal epithelial cells of *Candida* spp. plays a principal role in its pathogenesis. Additionally, the adhesion of *Candida* to the cells of the host, host cell proteins, or competing microorganisms hinders and/or diminishes the effectiveness of the host’s defence mechanisms in clearing the infection [103,104]. *C. ciferrii* is able to adhere to human buccal epithelial cells more than *C. krusei*, *C. glabrata*, and *C. orthopsilosis*, both reference and clinical isolates. The production of hydrolytic enzymes is also crucial in the pathogenesis of *Candida* and depends on the species of the *Candida* source and site of infection [105]. *C. ciferrii* has been shown to have proteolytic activity comparable to that of *C. albicans*, the *C. parapsilosis* complex, and *C. tropicalis* [106]. Besides proteases, phospholipases play an active role in the invasion of host tissue by *Candida* by disrupting the epithelial cell membranes and allowing the hyphal tip to enter the cytoplasm [107,108]. However, contrary to other NAC species, phospholipase activity in *C. ciferrii* has not yet been proven [40]. Haemolysins are a class of proteins defined by their ability to lyse red blood cells (RBCs), but they also exhibit pleiotropic functions [109]. Haemolysin, utilising the iron contained in the haemoglobin, activates the complement to opsonize the surface of RBCs [110]. This leads to host RBC damage and, thus, favours hyphal invasion in systemic candidiasis [111]. It has been reported that *C. ciferrii* exhibits haemolytic activity comparable to that of *C. albicans* and other NAC species [40]. The biofilm formation on mucosal or abiotic surfaces is another key virulence factor in *Candida* spp. In fact, *Candida* cells, which are embedded within biofilms, are more resistant to the host defence mechanisms and to the currently available antifungal drugs than their counterparts represented by planktonic cells. Biofilm formation on implant devices poses a serious problem to public health because it can lead to therapeutic failure [112,113,114,115,116]. Regarding the ability of *C. ciferrii* to produce biofilm, conflicting data are reported in the literature. In 2010, Seker et al. [117] reported that *C. ciferrii* was not able to produce biofilm in in vitro models. On the other hand, Leite et al. [118] evaluated the biofilm formation of *C. ciferrii* on catheter discs’ surface and demonstrated that during the first 24 h of incubation, *C. ciferrii* showed sparse yeast cells typical of the initial stage of the biofilm formation, but upon 48 h of exposure, a mature biofilm on the disc surface could be detected. Considering that the *S. ciferrii* complex is able to produce true hyphae, pseudo-hyphae, biofilm, proteases, and haemolysin, we suppose that it has a high potential to be considered as a human pathogen despite that clinical relevance infections are more frequent in immunocompromised subjects or in peculiar niches such as immune-privileged sites or skin appendages. Moreover, the high isolated rate of the *S. ciferrii* complex from the ear is extremely intriguing, as it is the same anatomical site of the isolation of *Candia auris*, despite the lacking in thermotolerance mechanisms in the *S. ciferrii* complex. Thus, the aforementioned mechanisms of virulence may play a crucial role in *C. ciferrii* pathogenicity, making this fungus a serious public health threat.

### 3.8. S. ciferrii Complex as Colonizer in Humans

Human skin is commonly colonised by a variety of fungal species. Fungi colonize epithelial surfaces of the human body in a mutualistic interaction with the host and other microorganisms of the skin microbiota. These interactions are regulated by multiple factors, including host physiology, immunity, and nutrient competition. Among fungi that colonize the skin and mucosal surfaces, *C. albicans* can be present both as a commensal or pathogenic microorganism [119]. However, host defence mechanisms at the skin barrier, involving residential non-immune cells and immune cells specifically recruited to the site of infection, are very efficient in counteracting the tissue invasion by *Candida.* Although there are a few case reports of skin and mucosal infections due to *C. ciferrii*, the *S. ciferrii* complex can also be considered a commensal microorganism of the human skin. Juavang et al. [120] reported a retrospective study where 184 *Candida* strains were isolated from July 2017 to July 2018 in the Philippines. In that study, the *C. ciferrii* complex represented 10% of the total isolates. The majority of *C. ciferrii* strains were isolated from respiratory and urinary specimens, in elderly patients (over 55 years old) with comorbidities. As to the source of *C. ciferrii*, it can be assumed that the infections may be the result of the translocation of the patient’s own microbiota [120]. These data are interesting based on the evidence that pulmonary and genital infections, due to the *S. ciferrii* complex, represent a small percentage of all human infections correlated with this species of *Candida* [29,34,35,70]. The genus *Candida* has been considered historically as a human commensal microorganism with a relatively low pathogenic potential. However, the status of *Candida* as an innocent bystander has recently been widely questioned by both clinical observations and animal experimentation [121], and the final word on the importance of *Candida* in the respiratory tract has not yet been said [120]. Foroozan et al. [122], studying the prevalence of fungal colonisation and/or infection at the ulcer site and the surrounding skin of lower extremities in a group of patients living in the south of France, found that *Candida ciferrii* was occasionally present in the surrounding skin but never in the ulcerative lesions. Interestingly, the study underlines how *C. ciferrii* has historically spread in specific geographical areas, also in consideration of the previous French studies reported by De Gentile et al. [13,26]. Based on this observation, we believe that the *S. ciferrii* complex could be a colonizer of the skin in humans and, occasionally, could be found as a transient fungus in the upper respiratory or in urogenital tracts in contrast to other NAC species that can colonize these apparatuses. Moreover, no data concerning the *S. ciferrii* complex in the human gut mycobiome or clinical disease have been reported, and we hypothesize that, in contrast to other NAC, the *S. ciferrii* complex does not have a trophism for the gastrointestinal tract despite that it is able to produce biofilm and is able to adhere to human buccal epithelial cells more than other NAC. In conclusion, as per Foroozan et al. [122], who did not find any association between ulcers and the direct presence of the *S. ciferrii* complex, we suppose that the *S. ciferrii* complex may be part of the resident skin microbiota although its exact role remains to be defined (Table 5).

### 3.9. S. ciferrii Complex in the One Health Perspective

Human activities have significantly impacted the environment and influenced the climate and the earth’s temperature with serious consequences for us, animals, plants, and microbial life on earth [123]. The inappropriate use of fungicides as well as several disruptive practices in farming, pisciculture, deforestation, and natural habitat destruction have contributed to the emergence of the multi-drug-resistant *C. auris* and other drug-resistant NAC species [124]. Almost all investigations carried out on nosocomial outbreaks of *C. auris* have concluded that this fungal pathogen was acquired from exogenous sources rather than from the patient’s endogenous microbiota. This finding suggests that despite the fact that no environmental reservoir has been clearly proven to date, it is more likely that *C. auris* may have its own environmental niche [125]. As for *C. auris*, other *Candida* spp. are not exclusively human pathogens or commensals since they have also been found in animals and several plants [126]. In this view, an adequate One Health approach is required to monitor environmental niches to track the emergence of cryptic pathogens. Previously largely ignored, the *S. ciferrii* complex has gained our attention only in recent years; therefore, it is not surprising that our knowledge of this fungus is still incomplete. The epidemiology of the *S. ciferrii* complex is currently being studied using molecular tools, but more research is needed. As the *S. ciferrii* complex has been recognised as a problematic pathogen in superficial and systemic candidiasis due to its drug resistance, a better understanding of host risk factors, mode of infection, and fungal-associated virulence determinants is of great importance. To date, the niche of the *S. ciferrii* complex has not yet been identified. The first reported strains of *C. ciferrii* were recovered from the neck of a cow, followed by another isolation from a wooden pole in a cow shed and then from a bovine placenta [15]. Moreover, in 2013, *C. ciferrii* was found on the body surface and jaws/pharynx of *Hirudo verbana* leeches in Poland [127], despite no human infections having been reported in that country. In 2000, Kano et al. [42] isolated in Japan the first *S. ciferrii* strain from a cat suffering from feline otitis followed by another case in 2014 in Brazil [128]. In 2006, Ullao et al. [129] reported that *Candida ciferrii* was the most common yeast species isolated in bat guano in Mexico. *S. ciferrii* was considered an environmental yeast, as it was found in vegetable as well as in animal materials, confirming previous reports [130]. In addition, in 2013, *S. ciferrii* was isolated on the skin and mucous membranes of English full-blood mares, indicating that these yeasts could be part of the cutaneous microbiota of horses [131]. The latest data regarding the isolation of *C. ciferrii* from animals and the environment document the presence of *C. ciferrii* in the gastrointestinal tract of alpacas in Poland [132] (Figure 7).

Environmental isolates of the *S. ciferrii* complex are found worldwide, suggesting that this fungus may be an inhabitant of the soil and marine aquatic environments [133]. Interestingly, the *S. ciferrii* complex was found in the soil worldwide and in different climate conditions. Moreover, *C. ciferrii* was isolated from the soil near Pretoria in South Africa, which is noteworthy because no human cases of infection have been reported in Africa [23]. Then, *C. ciferrii* was isolated in the soil taken from the farm of a goose farmer in Périgord, France. This is interesting because the first cases of *C. ciferrii* infections were described in France in 1991 [13,57]. In the same year, *C. ciferrii* was also found in an Antarctic dry valley [134], and six years later, Spencer et al. [135] found it in the pods of algarrobo trees (*Prosopis* spp.) in northwest Argentina. Moreover, Schroeder et al. [136], by evaluating the community composition and diversity of Neotropical root-associated fungi in trees, reported the presence of the *S. ciferrii* complex. As described in the literature, two cases of a *S. ciferrii* complex associated with candidemia were documented in Mexico [29,31]. In another report, a *S. ciferrii* complex was found in three Panamanian rainforests, supporting the hypothesis of the *S. ciferrii* complex as a soil and plant inhabitant [137]. In 2018, the *S. ciferrii* complex was reported to be part of the seagrass-associated fungal communities following Wallace’s line [138]. Islands possess peculiar features due to their geographical seclusion and resident biodiversity, making them an ideal setting for formulating and examining fundamental evolutionary and ecological principles or hypotheses [139]. Zheng et al. [140], by examining the soil fungal communities in 18 oceanic islands in Asia, including Hong Kong, found the presence of the *S. ciferrii* complex. Ettinger et al. [141], by exploring the mycobiome of the aquatic environment in the Japanese islands, identified the *S. ciferrii* complex as a part of the *Zostera marina* mycobiome. As reported above, only one case of *S. ciferrii* infection has been described in Japan, but no correlation was found with seawater or eelgrass contact [18]. Similarly, to the previous description of *Candida ciferrii*, *Candida mucifera* was described for the first time in 1988, from a frog liver near Manaus, Brazil [142]. In 2023, *C. mucifera* was found in a pit mud of Chinese strong-flavour liquor, in China [143]. Intriguingly, *Candida mucifera* human infections have only been reported in China [25]. Data regarding the isolation of *S. allociferrii* strains from animals or the environment have not yet been reported and only 12 strains have been isolated from humans. In summary, the literature data seem to support the hypothesis that the species of the *S. ciferrii* complex colonize both the environment and animals and, in some circumstances, can be pathogenic for humans. However, since this fungal pathogen can be transmitted from animals to humans, caution must be used whenever handling infected animals. Moreover, Brilhante et al. [144] reported a high rate of azole-resistant *Candida* spp. (*Debaryomyces*, *Meyerozyma*, and *Trichomonascus*), including two *C. ciferrii* strains, in an aquatic environment in Brazil, which may represent a human health risk. Intriguingly, in this work, the authors demonstrated for the first time the involvement of efflux pumps in the azole resistance of *Candida* spp. isolated from environmental sources [144]. Castelo-Blanco et al. [145] evaluated the azole resistance rate in humans and animals in Brazil, demonstrating a high prevalence of azole-resistant rates, especially for fluconazole and itraconazole, in *C. albicans* and NAC species. Moreover, they compared the epidemiological cut-off values in human and animal isolates, highlighting the existence of stronger selective pressures in animal-associated niches than in their human counterparts. Altogether, these data support the current theory on the existence of an environmental route for developing drug resistance in fungi and highlight the importance of the One Health approach to control the spread of azole resistance in fungal pathogens [145]. The fungal kingdom’s breadth and diversity ensure an endless reservoir of new and old variants of pathogens that show a prompt evolution when exposed to antifungal molecules, suggesting that in a changing environment, human health will always be inextricably linked to the complex ecology of fungal communities, either commensal or environmental, in a One Health perspective. Similarly, our concurrent need to control fungal disease in agricultural environments and in the clinic necessitates integrated responses. Pathogenic fungi are widely vectored, both actively and passively, so combating antifungal resistance requires a global coordinated response [146]. Considering the fact that microorganisms play a fundamental role in maintaining life on earth, the answers to the challenges of our time lie in a holistic global vision that recognizes the intimate connection between human health and the health of earth’s ecosystems.

## 4. Discussion and Conclusions

In October 2022, the WHO released the first-ever list of fungal “priority pathogens” that represent the greatest threat to public health, to guide research and public health action to strengthen the global response to fungal infections and antifungal drug resistance [147]. Various factors have contributed to the emergence of drug resistance in fungal pathogens in recent decades, such as environmental conditions, virulence determinants, and changes in gene expression [148]. Multiple studies have shown that the annual incidence of *Candida* spp. bloodstream infections is between 1.2 and 26 cases/100,000 individuals across the world. Only seven percent of them are community-acquired, and the remainder are healthcare-associated infections [149]. Candidemia epidemiology in Europe currently relies on individual efforts of committed researcher groups in the field of clinical mycology and microbiology. A meta-analysis study performed by Koehler and coll. [65] in 2019 investigated the available data about candidemia in Europe. Results of this investigation highlighted the existence of considerable differences among different clinical groups and European regions and over time [65]. However, a pan-European effort is clearly missing. It is needed to fill the gaps in our understanding of the epidemiology of candidemia, its associated drug resistance, and species shifts. A lethal case of *S. ciferrii*-complex-associated candidemia was reported in Austria, but no other systemic infections were reported in Europe, whereas great attention should be paid to developing countries. A prospective study conducted in 2015 in 27 ICUs in India showed a mean incidence of 6.51 cases of ICU-acquired candidemia per 1000 ICU admissions with an overall death rate of 35–75 percent [125]. It has been estimated that 14.3 million patients per year are admitted to ICUs in India. Blood culture sensitivity is only ≈40 percent for invasive candidiasis (and lower for intra-abdominal candidiasis). Further, the antifungal agents, including fluconazole and echinocandins, substantially reduce the yield from a blood culture. Therefore, it is probable that the actual number of cases of invasive and intra-abdominal candidiasis in ICUs in India exceeds 200,000 cases, resulting in ≈100,000 deaths per year. Bloodstream infections caused by *Candida* spp. outside ICUs are twice as common as in ICUs, and so in India > 600,000 persons each year are estimated to have invasive candidiasis. Inappropriate use of fungicides as well as several disruptive practices in farming, pisciculture, deforestation, and natural habitat destruction most likely have contributed to the emergence and spread of the multi-drug-resistant *C. auris* [150]. Interestingly, data from the literature show that *S. ciferrii* complex blood infections have been reported in developing countries such as India, Brazil, and Mexico and isolated cases in Taiwan (China) and Turkey. No correlations have been found with sex, age, immunocompetence, sensitivity to antifungals, and patients’ outcomes. Further studies are necessary to better understand the causes of death related to systemic infections by the *S. ciferrii* complex. There is little information on the prevalence of secondary fungal infections in clinical cases of chronic suppurative otitis media (CSOM). Otomycosis is sometimes diagnosed as a purely bacterial infection that is treated with antibiotics. In such cases, it is not possible to achieve complete healing due to the concomitant presence of a fungal agent. As above-mentioned, *Aspergillus*, *Candida*, and *Penicillium* spp. represent the most common fungal agents implicated in otomycosis [54,151]. In this review, we have underlined how the *S. ciferrii* complex mainly causes auricular infections, with complete recovery in all cases, independently from antimycotic susceptibility profiles, age, sex, and immunocompetence of patients, as well as geographical location. Moreover, all the cases are reported in young (25–38 years old) and immunocompetent patients. *Candida* pneumonia can be a primary infection of the lungs or represent a disseminated *Candida* infection associated with risk factors, such as long-term antibiotic use, haematologic malignancy, or severe immunosuppressive status [152]. According to our literature review, only two cases of pulmonary *S. ciferrii* complex infection have been reported in females with diabetes and both recovered completely in India and Turkey [29,38], supporting the hypothesis that the *S. ciferrii* complex rarely causes pulmonary infections. Although data on the epidemiology and antifungal susceptibility of *Candida* species infections in Europe and America are increasing, missing data in Eastern and Middle Eastern countries remain a reality, especially for non-life-threatening infections. The analysis of literature data reveals that in Malaysia and Bangladesh, the majority of VVCs are frequently due to *C. albicans*, but also to NAC such as *C. glabrata*, *C. parapsilosis*, *C. tropicalis*, and *C. krusei*. Among them, the *S. ciferrii* complex makes its own entry into local epidemiology as a cause of VVC, showing a high rate of resistance to itraconazole in addition to fluconazole [153]. In parallel to Malaysian data, *S. ciferrii*-complex-related VVC exhibits a high azole and AmB resistance rate but few data are currently available [34]. As for pulmonary infections, a few reports have documented VVC due to the *S. ciferrii* complex and further large-scale prevalence investigations should be performed to study the rate of itraconazole resistance and to conduct the best antifungal treatment in clinical practice. As reported above, ocular infections related to the *S. ciferrii* complex occur mainly in immunocompetent, female patients after iatrogenic procedures, supporting the hypothesis that they represent healthcare-associated infection. Furthermore, similarly to infections of the skin appendages and ears, the *S. ciferrii* complex seems to prefer external body areas at lower temperatures, perhaps due to a lack of thermotolerance mechanisms, as also reported in in vitro studies demonstrating that its growth is inhibited at 42 °C [154]. While dermatophytes remain the primary cause of OM, Candida species are emerging as important pathogens. Several factors contributing to OM have been characterised. Among them, there are defects in local immunity at the nail level, which can be due to traumatic lesions, psoriasis, lichen planus, corticosteroid use, prolonged exposure to moisture, and exposure to chemical agents [58]. Onychomycosis is the second most prevalent disease related to the *S. ciferrii* complex, after CSOM. In all the reported cases, except one, all the patients presented a full recovery and response to treatment despite the high prevalence of azole resistance. Moreover, most cases are reported in France, in line with the environmentally isolated strains in this country, supporting the hypothesis that the *S. ciferrii* complex is a soil and water inhabitant that can colonize or cause diseases in humans and animals, compelling close surveillance. An adequate One Health monitoring of environmental niches is required to track the emergence of similar pathogens. Interestingly, no data concerning gastrointestinal and nervous infection related to the *S. ciferrii* complex have been described. Although the mechanisms underlying these observations are still unclear and will need further studies, we suspect that the trophism of the *S. ciferrii* complex is mainly cutaneous such as for other species of *Candida*. However, we hypothesise that the *S. ciferrii* complex cannot colonize GIT and nervous systems due to the lack of thermotolerance mechanisms. The main limitation of our review is related to the small number of cases reported in the literature that should be implemented to create predictive algorithms for subjects at risk of *S. ciferrii* infection. Furthermore, the articles that we have analysed, according to the inclusion and exclusion criteria, lack some information, such as the immunological status, gender, and age of the subjects reported. Also, the time from the hospitalisation to diagnosis and the time from the diagnosis to medication are lacking; therefore, the cause of death could not be further discussed in all the cases. In addition, data on antifungal susceptibility testing and changes in the MIC cut-off are often partial or missing. Moreover, it should be noted that data on antifungal drug resistance began to be available in 1991, as well as the knowledge of the prescribed therapies. Furthermore, the genus identification of the *S. ciferrii* complex was performed in 2002, but only in 2015 in clinical practice. Besides *C. auris*, in recent years, the *Stephanoascus ciferrii* complex has been another emerging multi-drug-resistant NAC species. It seems that *S. ciferrii* complex infections are not associated with high mortality rates as this fungus mainly causes local mycoses such as ear and eye infections. However, a few cases of candidemia by the *S. ciferrii* complex were reported in severely immunocompromised individuals. The low susceptibility to currently available antifungal drugs is a rising concern, especially in NAC species. In this regard, a high rate of resistance to azoles, and more recently also to echinocandins, has emerged in the *S. ciferrii* complex, posing a serious clinical problem. The development of multidrug resistance among NAC species has become a global health threat. This is due to the difficulty in managing infections caused by drug-resistant microorganisms that very often are associated with high mortality rates and clinical failure. Thus, rapid and correct identification is extremely important to implement appropriate treatments aimed at limiting the spread of emerging drug-resistant fungal pathogens.

## Figures and Tables

**Figure 1 jof-10-00294-f001:**
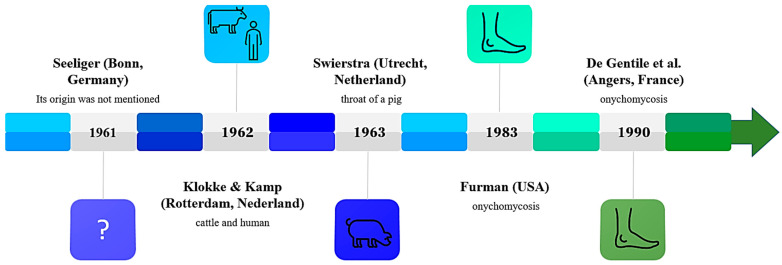
*C. ciferrii* timeline from 1961 to 1990 [13,15,16], when its role as a pathogenic yeast was established.

**Figure 2 jof-10-00294-f002:**
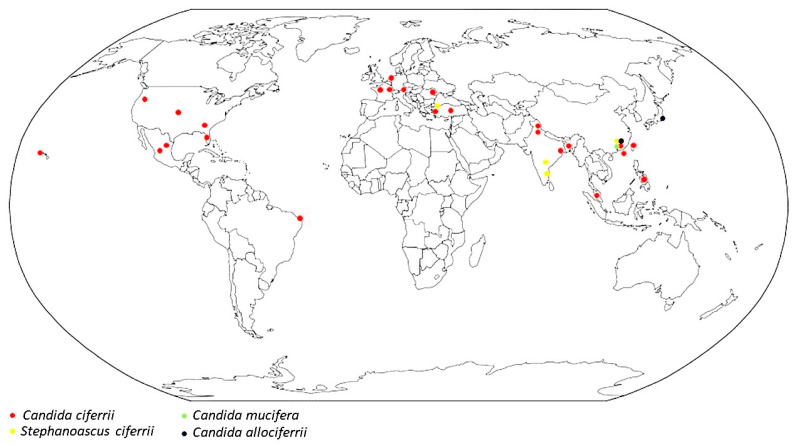
Geographic distribution of *Stephanoascus ciferrii* complex infections.

**Figure 3 jof-10-00294-f003:**
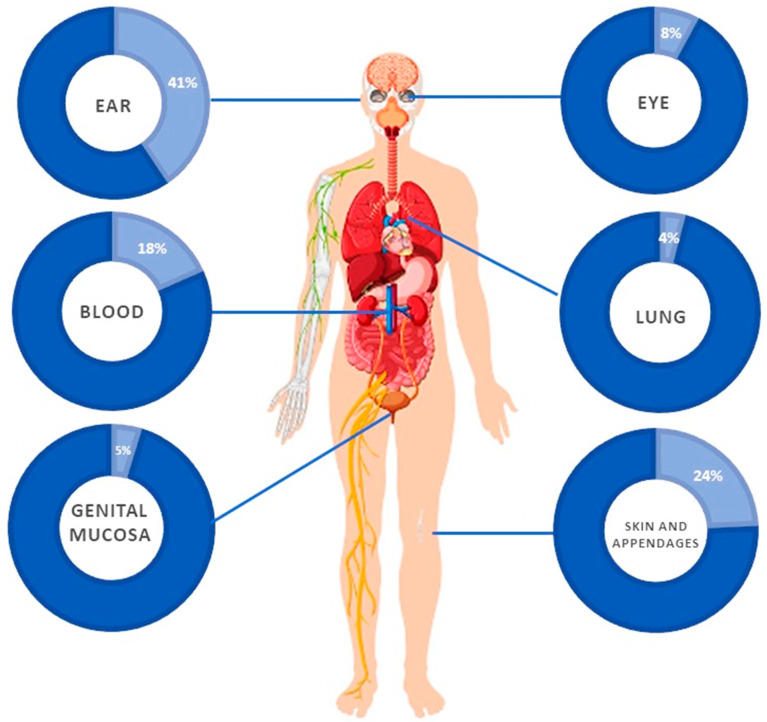
*Stephanoascus ciferrii* complex in human infections.

**Figure 4 jof-10-00294-f004:**
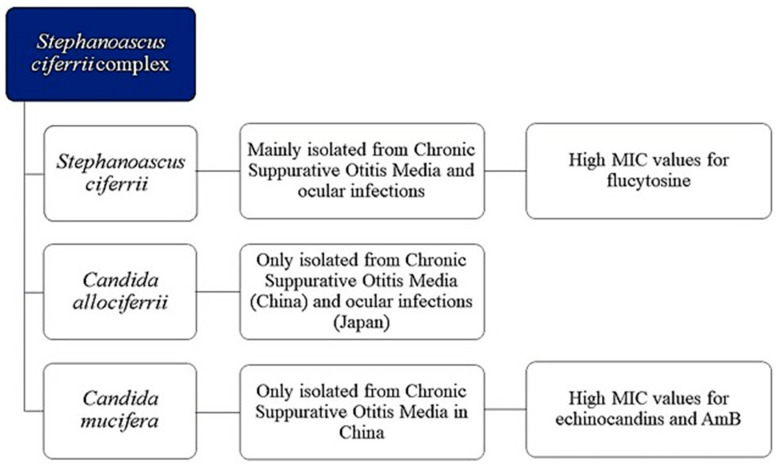
*S. ciferrii* complex-related diseases and their pattern of antifungal sensitivity. As 18S rRNA gene sequencing generated a new classification in 2002, previous studies were excluded, and only the period from 2003 to 2023 was considered. From the analysis of the literature, it emerges that *C. mucifera* is only described in China and has been isolated from CSOM. *C. allociferrii* has been described in both China and Japan. Furthermore, *C. mucifera* tended to have a higher MIC value than *C. allociferrii* for echinocandins and AmB. *S. ciferrii* tended to have higher MICs for flucytosine.

**Figure 5 jof-10-00294-f005:**
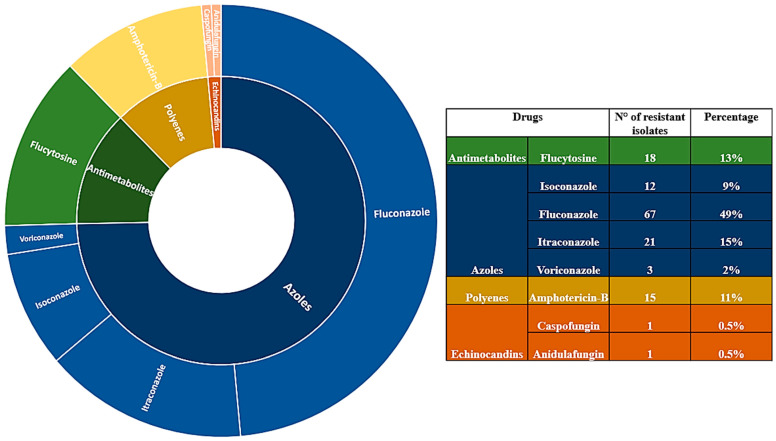
Resistance rates of *S. ciferrii* complex isolated strains. Only the cases reported after 1991 were considered, due to the lack of drug resistance data in the earlier period. It can be noted that, since 2001, resistance to azole drugs predominates, especially fluconazole, followed by resistance to antimetabolites (flucytosine).

**Figure 6 jof-10-00294-f006:**
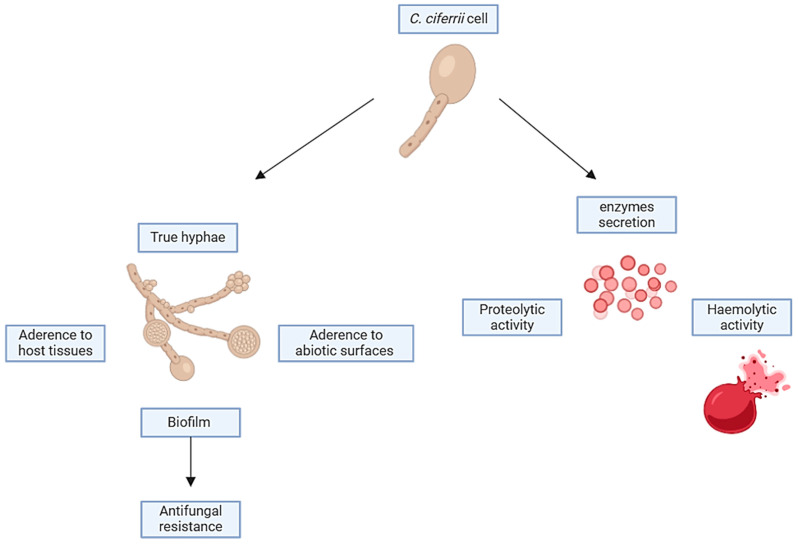
Virulence factors of *S. ciferrii* complex. Created with BioRender.com (accessed on 23 December 2023).

**Figure 7 jof-10-00294-f007:**
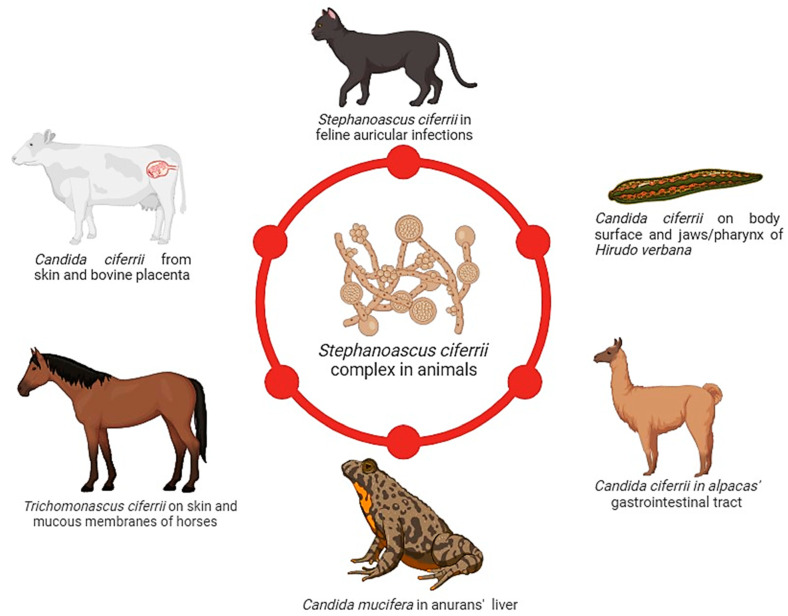
*S. ciferrii* complex strains isolated in animals. The *S. ciferrii* complex seems to be part of cutaneous and mucosal microbiota in horses and cows, the gastrointestinal tract of alpacas, and *Hirudo verbana*, while only two cases of feline auricular infection were described. The figure was created using BioRender.com (accessed on 23 December 2023).

**Table 1 jof-10-00294-t001:** Clinical description of patients with *S. ciferrii* complex infection collected from 1983 to 2023.

Reference	Years	Clinical Presentation	Sex	Ethnicity	Age (Years)	Immunity State	Underlying Conditions	Initial Symptoms/Signs	Outcomes	Treatment	Resistance (MIC)	Identification	Country
Furman and Ahearn [16]	1983	Otomycosis	Male	NA	25	NA	NA	Oesophageal pain. Headaches. Ear pressure	Recovery	1% hydrocortisone; 2.0% acetic acid solution	NA	Morphological and physiological identification	USA (Kansas)
		Onychomycosis	Female	NA	NA	NA	NA	NA	NA	NA	NA		USA (Hawaii)
		NA	Male	NA	NA	NA	NA	NA	NA	NA	NA		USA (Oregon)
		Gangrenous foot	Female	NA	68	NA	NA	NA	NA	NA	NA		USA (Geogia)
		*Tinea pedis*	NA	NA	NA	NA	NA	NA	NA	NA	NA		USA (Florida)
De Gentile et al. [13]	1991	Toenail onyxis	NA	NA	52	NA	NA	NA	NA	NA	NA	ATB 32C (API system)	France
	1991	Toenail onyxis	NA	NA	69	NA	NA	NA	NA	NA	NA		France
	1991	Toenail onyxis	Male	NA	88	NA	Auricular fibrillation. Extensive cutaneous ulceration	Ingrowing toenails	NA	NA	Flu; Flucy (microdilution)		France
	1991	Onychomycosis	Male	NA	70	NA	Arteriopathy. Alcoholic. Smoker. Venous insufficiency with large, perforated ulcers	Onychopathy	NA	NA	NA		France
	1991	Toenail onyxis	NA	NA	84	NA	Trophic disorders. Nummular eczema. Farmer	NA	NA	NA	NA		France
	1991	Toenail onyxis	Male	NA	75	Immunocompromised	Diabetes. Rheumatism. Valvulopathy. Ulceration legs. *Tinea pedis*	*Tinea pedis*. Onyxis. *Tinea cruris*	NA	NA	NA		France
De Gentile et al. [26]	1995	Onychomycosis	12 patients	NA	35–95 (mean: 74)	NA	Trophic disorders	NA	NA	NA	Iso; Flu; Itra	API 20C AUX; ID 32C	France
Gunsilius et al. [27]	2001	Invasive candidiasis	Male	Caucasian	62	Immunocompromised	Acute myeloid leukaemia (AML, FAB-M1)	Fever. Erythematous skin papules	Exitus	Liposomal AmB (5.3 g)	Flu (>64 µg/mL)	API 20C AUX	Austria
Cheng et al. [28]	2004	Candidemia	Male	NA	NA	Immunocompetent	NA	NA	Recovery	NA	Flu (16 µg/mL)	API-32C	China (Taiwan)
Agin et al. [19]	2011	Candidemia	Male	Turkey	8	Immunocompetent	Down Syndrome. Cerebral palsy	Respiratory distress. Cough. Fever. Decubitus ulcer on the right gluteal region	Exitus	Lipid Complex AmB	AmB (1 µg/mL); Flu (64 µg/mL); Caspo (32 µg/mL); Anidu (32 µg/mL)	Vitek2	Turkey
Saha et al. [29]	2013	Pneumonia	Female	NA	55	Immunocompromised	Bidi smoker. COPD. Diabetes.	Dyspnoea. Mucopurulent cough	Recovery	Intravenous liposomal AmB (150 mg daily) was initiated and changed to oral Flu (150 mg daily) after 4 days	No	Vitek2	India
Daielescu et al. [20]	2014	Endophthalmitis	Female	East Europe	57	Immunocompetent	Cataract	VA decreased after cataract surgery	Recovery	50 µg/0.1 mL and then 250 µg/0.1 mL intravitreal Caspo	Flu Vori AmB	Vitek2	Romania
Soki et al. [17]	2015	Infraorbital abscess	Male	NA	79	Immunocompromised	Enucleation of choroidal melanoma	Infraorbital abscess	Recovery	Topical AmB 0.1%	Flu (64 μg/mL)	18S rDNA, ITS1-26S rDNA D1/D2 region sequence	Japan
Vilanueva-Lozano et al. [30]	2016	Candidemia	Female	Mexican	30	Immunocompromised	Crohn’s disease. *Mycobacterium bovis* disease	Headache. Fever. Disorientation	Recovery	Posa	Flu (32 µg/mL)	Vitek2	Mexico
Dave et al. [31]	2018	Endophthalmitis	Female	NA	50	Immunocompetent	Chronic post-cataract surgery endophthalmitis	IOL plaque	Recovery	Oral keto tablet (200 mg) twice/d with topical Nata 5% eye drops and intravitreal injection of AmB (5 mg/0.1 mL)	AmB	Vitek2	India
	2018	Endophthalmitis	Female	NA	59	NA	Chronic post-cataract surgery endophthalmitis	Mutton-fat keratic precipitates. Vitritis. AC membranes	Recovery	Intravitreal AmB injection (5 mg/0.1 mL)	NA	Vitek2	India
	2018	Endophthalmitis	Female	NA	79	NA	Chronic post-cataract surgery endophthalmitis	Panuveitis	Recovery	Intravitreal AmB and oral keto (200 mg) 2 times/day	NA	Vitek2	India
	2018	Endophthalmitis	Male	NA	66	NA	Chronic post-cataract surgery endophthalmitis	Hypopyon. Low-grade vitritis	Recovery	Oral Itra tablet (100 mg) twice/d with topical Nata 5% eye drops, and intravitreally, an injection of AmB (5 mg/0.1 mL)	NA	Vitek2	India
Bansal et al. [32]	2021	Endophthalmitis	Female	Asian Indian	26	Immunocompetent	Uncomplicated vaginal delivery	VA decreased	Recovery	Flu (3 months)	No	Vitek2	India
Guo et al. [25]	2021	Otomycosis	17 males; 15 females	NA	Mean: 36	NA	NA	CSOM	NA	NA	13 (40.6%) Flucy 32 (100%) Flu 32 and 256 μg/mL	ITS region rDNA sequence	China
Robles-Tenorio et al. [33]	2022	Onychomycosis and candidemia	Male	Mexican	58	Immunocompromised	Diabetes mellitus type 2 (DM2)	Cough. Headache. Dyspnoea	Exitus	AmB	Flu	Vitek2	Mexico
Sathi et al. [34]	2022	9 candidemia; 2 ocular diseases; 1 VCC	12 patients	NA	NA	NA	NA	NA	NA	NA	Blood samples: 6 resistant to Flu, Itra; 1 resistant to Vori; 5 resistant to Amb. From eye: 1 resistant to Flu, Itra, Cotri. From HVG: 1 resistant to Flu, Itra, Vori, AmB		Bangladesh
Ng et al. [35]	2013 (cases from 2000)	Vaginal candidiasis	3 females	NA	NA	NA	NA	NA	NA	NA	NA	ITS1-5.8S-ITS2 region	Malaysia
Romald et al. [36]	2019	CSOM	Female	Indian	57	Immunocompetent	Post-mastoidectomy	Ear pain, profuse ear discharge, and hard of hearing in left ear for past three months	Recovery	Oral Vori and topical Clotri	Flu > 32 μg/mL Itra ≥ 16 μg/mL Vori < 0.025 μg/mL AmB ≥ 16 μg/mL Posa ≤ 0.025 μg/mL Caspo ≤ 0.025 μg/mL	ITS1 4	India
Demiray et al. [37]	2015	Candidemia	Female	NA	Newborn, 23 days	NA	Neonatal diaphragmic hernia	Systemic mycosis. Cutaneous lesions. Respiratory distress	Recovery	Flu (1 × 21 mg/d)	AmB ≤ 0.25 µg/mL Flucy ≤ 1 µg/mL Flu ≤ 1 µg/mL Vori < 0.12 µg/mL	API system	Turkey
Papìla et al. [38]	2016	Septic pulmonary embolism	Female	NA	70	NA	Patient with type 2 DM	Altered consciousness. Speech disorder. Vomiting. Loss of power in the left. Confusion. Pupillary isocoria. Deep tendon reflexes (DTRs). Direct light reflex (DLR) (++/++). Left hemiplegia	Recovery	Anidu (200 mg daily loading dose and 100 mg daily maintenance dose) for 4 weeks; treatment was discharged with 400 mg Flu (2 weeks)	NA	Vitek2	Turkey
Chan et al. [39]	2015	Granular myringitis	3 males; 1 female	NA	Mean: 38 Range:33–42	3 immunocompetent; 1 immunosuppressed	3 no comorbidities. 1 nasopharyngeal carcinoma, diabetes mellitus, and hyperlipidaemia	Obstructive sensation and/or pruritis in ears. Pustular otorrhoea. Otalgia	Recovery	Responses to treatment with irrigation and topical antifungal cream are generally good	Flu; AmB; Flucy	Vitek2 YST ID Card system. API 20C AUX; MALDI-TOF MS; ITS	China (Hong Kong)
Vieira de Melo et al. [40]	2019 (blood collection from 2011 to 2015)	Candidemia	NA	NA	NA	NA	NA	NA	NA	NA	NA	NA	Brazil
Capoor et al. [41]	2015	Fungemia	Female	NA	30	Immunosuppressed	AML Fluconazole prophylaxis	NA	Expired	NA	Resistant to AmB, Flu, Itra	API 20C AUX	India

AmB, Amphotericin-B; AC, Anterior Chamber; AML, Acute Myeloid Leukaemia; Anidu, Anidulafungin; Caspo, Caspofungin; Clotri, Clotrimazole; COPD, Chronic Obstructive Pulmonary Disease; CSOM, Chronic Suppurative Otitis Media; Flu, Fluconazole; Flucy, Flucytosine; IOL, Intraocular Lens; Iso, Isoconazole; Itra, Itraconazole; keto, Ketoconazole; NA, Not Available; Nata, Natamycin; MIC, Minimum Inhibitory Concentration; Posa, Posaconazole; VA, Visual Acuity; VVC, Vulvovaginal Candidiasis; Vori, Voriconazole.

**Table 2 jof-10-00294-t002:** Morphological characteristics of *S. ciferrii* complex colonies grown on different solid culture media.

Reference	Medium	Texture	Colour
Furman and Ahearn, 1983 [16]; Kano et al., 2000 [42]	SDA	Small; round; could be tough, raised, and wrinkled or smooth	Milky white or cream
Romald et al., 2019 [36]	TTZ	Dry, rough, wrinkled	Pink to white
	PDA	Rough	Non-pigmented
	OMA	Wrinkled	White
	CMA	Dry, rough, wrinkled	
	HiCrome™	Dry, wrinkled	Blue-white
Guo et al., 2021 [25]	CHROMagar™	Regular round	Blue centre and white edge

CMA, Cornmeal Agar; OMA, Oatmeal Agar; PDA, Potato Dextrose Agar; SDA, Sabouraud Dextrose Agar; TTZ, Tetrazolium Reduction Medium.

**Table 3 jof-10-00294-t003:** Studies in which *S. ciferrii* complex was misdiagnosed by biochemical or molecular methods.

		Misidentification		
Reference	Isolated Pathogen	Biochemical	Molecular	Proteomic
Balada-Llasat et al., 2012 [49]	*Candida ciferrii*		*Cryptococcus neoformans*	
Kumar et al., 2014 [44]	*Blastobotrys serpentis*	*Stephanoascus ciferrii*		
	*Blastobotrys proliferans*	*Stephanoascus ciferrii*		
Roehmel et al., 2015 [48]	*Arxula adeninivorans*			*Stephanoascus ciferrii*
Warren et al., 2017 [14]	*Cryptococcus laurentii*	*Candida ciferrii*		
A. Al-Haqqan et al., 2018 [45]	*Candida blankii*	*Candida ciferrii*		

**Table 4 jof-10-00294-t004:** Distribution of resistant isolates of *S. ciferrii* complex by specimen type. Only studies in which drug-resistant data were available have been included. Some isolates were resistant to more than one drug.

	N° of Resistant Isolates of *S. ciferrii* Complex	Flucy	Iso	Flu	Itra	Vori	AmB	Caspo	Anidu
Skin and appendages	22	1 (4.5%)	12 (54%)	14 (64%)	12 (54%)	0	0	0	0
Blood	17	0	0	12 (71%)	7 (41%)	1 (6%)	7 (41%)	1 (6%)	1 (6%)
Ear	41	17 (41%)	0	41 (100%)	1 (2%)	0	4 (8%)	0	0
Eye	7	0	0	2 (29%)	0	1 (14%)	3 (43%)	0	0
Lung	2	NA	NA	NA	NA	NA	NA	NA	NA
Genital mucosa	4	0	0	1 (25%)	1 (25%)	1 (25%)	1 (25%)	0	0

AmB, Amphotericin-B; Anidu, Anidulafungin; Caspo, Caspofungin; Flu, Fluconazole; Flucy, Flucytosine; Iso, Isoconazole; Itra, Itraconazole; NA, Not Available; Posa, Posaconazole; Vori, Voriconazole.

**Table 5 jof-10-00294-t005:** Studies reporting *S. ciferrii* complex as a colonizer in humans.

Reference	Years	Clinical Presentation	Sex	Ethnicity	Age (Years)	Immunity State	Underlying Conditions	Resistance (MIC)	Identification	Country
Foroozan et al. [105]	2011	NA	NA	NA	NA	NA	Low-extremity cutaneous ulcers	NA	NA	France
Juayang et al. [103]	2017–2018 (published 2019)	19 patients	NA	NA	17 cases > 55 years old; 2 cases < 55 years old	NA	NA	42.1% Fluco; 38.9% Voria	VITEK 2	Philippines

MIC data missing. Fluco, fluconazole; MIC, minimum inhibitory concentration; NA, Not Available; Vori, voriconazole.

## Data Availability

All data of this research are reported in the main text and Appendix A.

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
