# Peer review of "Stephanoascus ciferrii Complex: The Current State of Infections and Drug Resistance in Humans"

_jof, 2024, doi:10.3390/jof10040294_

Round 1

Reviewer 1 Report

I think this could be a good review. However, due to plagiarization this can not be accepted in its current state. 

In addition, I would like the authors to include any molecular characterization of this organism. For example, what is know about the genome size, is it haploid or diploid, does it use an alternate codon, are their any genetics, what about the sexual or is it asexual? Adding the inform would be informative for the nonexperts of the field.  If it isn't know that would also be useful.  Possibly, you can compare other candida strains in a table format. I would also rephrase the title.  It is confusing using complex and interactions next to each other. May be rephrase as "Stephanoascus ciferrii complex: The current state of infections and drug resistance in humans". You also need to describe what the S. ciferrii complex is within the abstract.

That authors need to fix the plagiarized lines. Provided are a few examples but there are more in the text. I would recommend going through the entire text and or running through a plagiarism checker.

Lines 68-73, lines 85-88, Lines, 155-160, Lines 283-287, Lines 549-553

Author Response

We would like to thank the Reviewers for taking the time to review the manuscript. We sincerely appreciate all valuable comments and suggestions, which helped us to improve the quality of the manuscript.
Here is a point-by-point response to the reviewers’ comments and concerns.

Major comments
I think this could be a good review. However, due to plagiarization this can not be accepted in its current state.
In addition, I would like the authors to include any molecular characterization of this organism. For example, what is know about the genome size, is it haploid or diploid, does it use an alternate codon, are their any genetics, what about the sexual or is it asexual? Adding the inform would be informative for the nonexperts of the field. If it isn't know that would also be useful. Possibly, you can compare other candida strains in a table format. I would also rephrase the title. It is confusing using complex and interactions next to each other. May be rephrase as "Stephanoascus ciferrii complex: The current state of infections and drug resistance in humans". You also need to describe what the S. ciferrii complex is within the abstract.

Dear Reviewer, thank you for your valuable comments. They have helped us to improve the manuscript. Please find our point-by-point response below.

A R : The title has been modified as you suggested to avoid confusion : « Stephanoascus ciferrii complex: The current state of infections and drug resistance in humans ». Regarding the molecular characterization of S. ciferrii, please see section 3.3 « Morphology and identification of S. ciferrii complex » where we reported the published data on the evolution, genome and phylogenetic analysis.

Detail comments
1)That authors need to fix the plagiarized lines. Provided are a few examples but there are more in the text. I would recommend going through the entire text and or running through a plagiarism checker.
Lines 68-73, lines 85-88, Lines, 155-160, Lines 283-287, Lines 549-553

1)Dear reviewer, thank you for the suggestion. All the main file has been checked for plagiarism and modified. Moreover, for English language editing the paper has been revised with the assistance of a native English speaker who is a competent technical writer.

2)Does the title describe the article's topic with sufficient precision, bearing in mind that it is a review article?
Please rephrase the title. It is confusing using complex and interactions next to each other. May be rephrase as "Stephanoascus ciferrii complex: The current state of infections and drug resistance in humans"

2)Dear Reviewer, thank you for the suggestion. The title has been modified as you suggested to avoid confusion: « Stephanoascus ciferrii complex: The current state of infections and drug resistance in humans »

3)Does the abstract/introduction provide a sufficiently clear description of the topic subject of this review?
You need to describe what the S. ciferrii complex is.
3)Dear reviewer, thank you for the suggestion. We have added a brief description of S. ciferrii complex in the abstract to clarify this aspect.

Reviewer 2 Report

The review article by Cosio et al, is extensively written and is an interesting read. The article describes in detail the research performed to date on Stephanoascus ciferrii. S. ciferrii is a non-albicans species of Candida causing mostly local and systemic fungal infections. The review article extensively covers, characteristics, diagnostics, treatment, virulence and drug resistance development in this novel Candida sp. The figures in this review article are good quality and supports the text. 

Here are my comments regarding the article:

1) From the evolutionary stand point, how is S. ciferrii related to other Candida sp. If known, please provide a description on it.

2) Are drug resistance mechanisms studied in clinical isolates of S. ciferrii? The authors mention in line 314 that echinocandin resistance has been desribed in clinical isolates... however, if the mechanisms of resistance has been studied in the clinical isolates of S. ciferrii, the authors should elaborate on it. Please clarify

3)line 362: "isolated from China in auricular samples (94,5%) -- did the authors mean 94.5%. Please clarify.

4) Fig 5: The each column in the table should have a header for better clarity.

5) Is whole genome sequencing available on S. ciferrii? Please elaborate on the text how S. ciferrii is genetically similar/different between the different with other Candida sp. 

Author Response

We thank the Reviewers for taking the time to review the manuscript. We sincerely appreciate all valuable comments and suggestions, which helped us improve the manuscript's quality.

Here is a point-by-point response to the reviewers’ comments and concerns.

Major comments
The review article by Cosio et al, is extensively written and is an interesting read. The article describes in detail the research performed to date on Stephanoascus ciferrii. S. ciferrii is a non-albicans species of Candida causing mostly local and systemic fungal infections. The review article extensively covers, characteristics, diagnostics, treatment, virulence and drug resistance development in this novel Candida sp. The figures in this review article are good quality and supports the text.

Dear Reviewer, thank you for your valuable comments. They have helped us to improve the manuscript. Please find our point-by-point response below.

Detail comments
Here are my comments regarding the article:
1)From the evolutionary stand point, how is S. ciferrii related to other Candida sp. If known, please provide a description on it.
1)Dear Reviewer, thank you for the suggestion. This point has been addressed in the section 3.3 « Morphology and identification of S. ciferrii complex » where we reported the published data regarding the evolution, genome and phylogenetic analysis of C. ciferrii.

2)Are drug resistance mechanisms studied in clinical isolates of S. ciferrii? The authors mention in line 314 that echinocandin resistance has been desribed in clinical isolates... however, if the mechanisms of resistance has been studied in the clinical isolates of S. ciferrii, the authors should elaborate on it. Please clarify
2) Regarding the mechanisms of drug resistance in S. ciferrii clinical isolates no detailed studies have been performed. However, we have added a description of the current knowledge regarding the closest Candida spp. to S. ciferrii to hypothesize the mechanism of resistance in the section « Treatment and resistance ».

3)Line 362: "isolated from China in auricular samples (94,5%) -- did the authors mean 94.5%. Please clarify.
3)It has been corrected as 94.5%.

4)Fig 5: The each column in the table should have a header for better clarity.
4) Dear reviewer, thank you for the suggestion. We have add headers to all figures.

5) Is whole genome sequencing available on S. ciferrii? Please elaborate on the text how S. ciferrii is genetically similar/different between the different with other Candida sp.
5) Dear reviewer, thank you for the suggestion. We have addressed this point in the comment 1. Please see the section 3.3 « Morphology and identification of S. ciferrii complex » the published data regarding the evolution, genome and phylogenetic analysis of C. ciferrii

Round 2

Reviewer 1 Report

The authors did not address the plagiarism issues. Modifying the plagiarized sentences by removing or adding a few words is still plagiarism and it is quite obvious. Thus, the manuscript in it's current state is still not written in your own words.

The authors have not fix their plagiarism issue! Thus, it hard to evaluate this manuscript based on originality of thought by the authors.  They might have removed or modified parts of the sentence but they still use phrases and/or order of how they were originally written. The authors need to use own words/thoughts instead of taking from others. There are many ways to construct a sentence/ideas without using someone's else's exact words, parts of phrasing, and/or sentence structure. 

Author Response

Dear Reviewer,

We thank you for taking the time to review the manuscript. We appreciate all valuable comments and suggestions, which helped us improve the manuscript's quality. As you can see in the attached report, the plagiarism detention rate is 17%. In detail, the plagiarism concerns mainly the authors' affiliation and the references with a very low percentage in the main text. Despite the low percentage rate of plagiarism detected, we have considered your comment to repeat a critical analysis of our manuscript. We would also like to underline that we have contributed with our own ideas/conclusions to the subject of our review.

Best regards,

Terenzio Cosio and Co-authors
